# Prototype-Guided Dual-Transformer Reasoning for Video Individual Counting

## ABSTRACT

Video Individual Counting (VIC), which focuses on accurately tallying the total number of individuals in a video without duplication, is crucial for urban public space management and densely-populated areas planning. Existing methods suffer from limitations in terms of expensive manual annotation, and the efficiency of location or detection algorithms. In this work, we contribute a novel Prototype-guided Dual-Transformer Reasoning framework, termed PDTR, which takes both similarity and difference of adjacent frames into account to achieve accurate counting in an end-to-end regression manner. Specifically, we first design a multi-receptive field feature fusion module to acquire initial comprehensive representations. Subsequently, the dynamic prototype generation module memorizes consistent representations of similar information to generate prototypes. Additionally, to further dig out the shared and private features from different frames, a prototype cross-guided decoder and a privacy-decoupling module are designed. Extensive experiments conducted on two existing VIC datasets, consistently demonstrate the superiority of PDTR over state-of-the-art baselines.

## CCS CONCEPTS

• **Computing methodologies → Computer vision problems**; **Computer vision tasks**.

## KEYWORDS

Video Individual Counting, Prototype Learning, Dual-Transformer

## 1 INTRODUCTION

With the ongoing increase in urban population density, effective crowd management and safety control in densely populated public areas are of utmost importance. Consequently, Video Crowd Counting (VCC) is developed to estimate the number of individuals in each frame, which has been widely used in various applications [7, 9, 40]. However, despite the fact that VCC has achieved amazing improvement, it fails to accurately count the total number of individuals appearing in a video, which follows a constraint that each pedestrian is counted only once. Therefore, a more challenging task termed Video Individual Counting (VIC) is proposed. Obviously, removing duplication counting and achieving accurate total enumeration from a video play as a critical issue in VIC.

*ACM MM, 2024, Melbourne, Australia*
© 2024 Copyright held by the owner/author(s). Publication rights licensed to ACM.
ACM ISBN 978-x-xxxx-xxxx-x/YY/MM
https://doi.org/10.1145/nnnnnnn.nnnnnnn

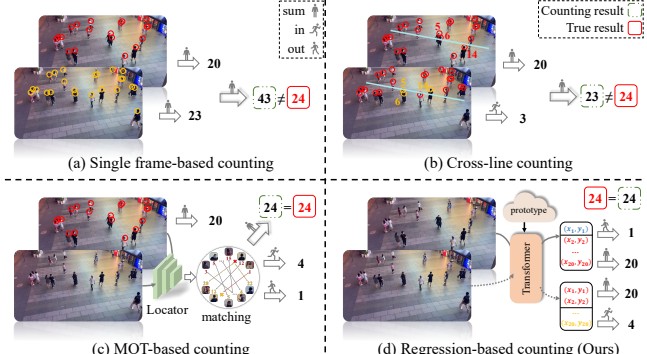

**Figure 1: Illustration of different VIC methods. (a) Single frame-based counting, adding the number of people detected in each frame as the total number. (b) Cross-line counting, counting the number of individuals crossing the virtual line as inflow. (c) MOT-based counting, using locators to generate head descriptors for different frames for feature matching. (d) Regression-based counting (Ours), mapping input frames into shared and private coordinate sets directly.**

Over the decades, VCC [1, 9, 42, 46] has witnessed significant advancements. Current methodologies generally fall into two categories: single frame-based counting and cross-line counting, neither of which is directly applicable to the VIC task. The single frame-based counting methods [11, 20, 29, 37, 52] inevitably lead to repetitive counting of the same target in adjacent frames, resulting in an inaccurate representation of the total count of people in a video, as shown in Fig. 1(a). The cross-line methods [4, 28, 47, 48] solely focus on counting the pedestrians crossing predetermined lines, disregarding pedestrians in stationary or intermittent motion states, as shown in Fig. 1(b). Additionally, the manual marking operation required by cross-line methods necessitates labor-intensive effort, and lead to prohibitive costs when dealing with highly dense conditions. Inspired by multi-object tracking (MOT) methods, VIC was initially defined by Han et al. [12] who proposed a new counting paradigm DRNet to tally the individuals in the initial video frame and increment this count with new entrants in subsequent frames. DRNet extracts head descriptors from density maps of adjacent frames and uses feature matching inference to infer inflow (the number of people entering the current frame) and outflow (the number of people who left the previous frame). Recently, Liu et al. [26] also proposed a weakly supervised method based on contrastive learning with group-level matching, to further reduce the reliance of MOT-based methods on trajectory labels. Unfortunately, these MOT-based methods (shown in Fig. 1(c)) heavily depend on the accuracy of density maps and descriptors.

In addition, the aforementioned methods all rely on convolutional neural network architecture and involve additional preprocessing operations. To reduce the reliance on these operations, Liang et al. [22] takes the lead in applying DETR [3] to crowd localization, thereby achieving an end-to-end regression localization. Despite the fact that this straightforward and efficient method is primarily suited for single image and cannot be directly applied to VIC, it prompts us to pose the hypothesis: *can video individual counting also be addressed using a uncomplicated regression approach?* Generally, the VIC task has higher difficulty and meets two key challenges: i) if the model can generate semantic and consistent representations of individuals across consecutive frames, it can effectively address the issue of target repetition in temporal individual counting; and ii) Given the continual motion state of target in a video, it becomes even more critical to thoroughly address the fusion of information pertaining to the same target across different frames.

Inspired by aforementioned analysis, we propose the Prototype-guided Dual-Transformer Reasoning (PDTR) framework, fully leveraging the advantages of prototype learning and dual-stream inference, as shown in Fig. 1(d). In particular, our PDTR is composed of four main components: i) multi-receptive field feature fusion module (MRF$^3$), ii) dynamic prototype generation (DPG), iii) prototype cross-guided decoder (PCD), and iv) privacy-decoupling module (PDM). Obviously, the sizes of targets in a frame are rich diverse for the sake of distances between targets and the camera. We propose the MRF3 to acquire and merge multi-scale representations. Subsequently, these representations are input into DPG to extract probability distribution of similar information across different frames, thereby directing the model's attention towards similarity and aiding in prototypes generation. Afterwards, a prototypes cross-guided mechanism is designed to efficiently train the transformer decoder to extract shared features from comprehensive representations of different frames, thereby inferring targets that present in both frames. Lastly, PDM is utilized to separate private features from each frame for complementary accounting.

The core contributions of this work are as follows:

- We propose a novel prototype-guided dual-transformer reasoning framework for VIC, which converts the feature matching process of conventional models to an end-to-end regression reasoning procedure. To the best of our knowledge, this represents *the initial endeavor* to employ Transformer in a dual-stream cross-guidance manner for VIC.
- A novel dynamic prototype generation module is deployed to bridge and mine consistency information from comprehensive representations of adjacent frames, assisting decoder in cross-generating semantic consistency features, thereby reasonably utilizing the motion information of targets between frames to reduce duplicate counting.
- Extensive experiments are conducted on two challenging benchmarks for video individual counting, which demonstrate: (a) the favorable comparison of our model with other state-of-the-art methods, and (b) the effectiveness of each module through ablation studies.

## 2 RELATED WORK

### 2.1 Video Individual Counting

Video individual counting (VIC), as it involves counting each person in a video only once, presents a higher level of complexity than frame-by-frame video crowd counting (VCC). One possible solution is the multi-object tracking methods [2, 23, 27, 30, 33, 34, 38], which detect and track multiple objects in consecutive frames of a given video, thereby achieving pedestrians counting. This scheme has made some research progress in VCC. Ren et al. [30] modeled the detection, counting, and tracking problems as a network flow problem. Sundararaman et al. [34] developed two frameworks for head detection and tracking, based on motion models and a color histogram-based re-identification module. However, multi-object tracking methods need to capture continuous trajectories of objects and consider frequent ID switching, they cannot effectively be applied to VIC. Motivated by these methods, Han et al. [12] innovatively converted VIC into a feature matching task by utilizing the descriptors of each located head in adjacent frames to match and determine pedestrian inflow. Liu et al. [26] proposed a novel baseline equipped with a newly designed group level matching soft contrastive loss. To our knowledge, these are the only two studies that completely focus on VIC. Unlike these methods, our approach reformulates the mainstream object matching process into a more reliable, prototype-guided similarity and difference reasoning procedure.

### 2.2 Prototype Learning

Prototype refers to the feature representations of instances within the same class [31]. Thanks to its exemplar-driven nature and simpler inductive bias [14], it holds significant potential across various tasks. In image classification, each prototype acted as a representation of a specific category, enhancing model performance through the regulation of inter-prototype distances [31, 43]. In semantic segmentation, prototype vectors represented the masked object features of support images and were utilized to search for pixel positions with similar features in the query image, thereby aiding in accurate target segmentation. [18, 39, 50]. In crowd counting, Huang et al. [15] established a weather bank for gathering various weather prototypes, and devised prototype loss to improve the adaptability under various weather conditions. Our study adopts the concept of prototypes to signify similarity and implements prototype aggregation within dual-transformer framework.

### 2.3 Visual Transformers

Transformer, initially proposed for modeling sequential data in machine translation [35], has recently demonstrated remarkable success in various tasks, including object detection [3, 5, 44], image recognition [8, 32], and semantic segmentation [49]. This success has led to increased interest in exploring transformer-based architectures for crowd counting [21, 22, 24]. For instance, Liang et al. [22] devised a novel approach by treating crowd localization as a regression task and developed an end-to-end transformer framework. Lin et al. [24] enhanced transformer by introducing learnable region attention and an instance attention mechanism, which effectively integrated global and local features to address scale differences.

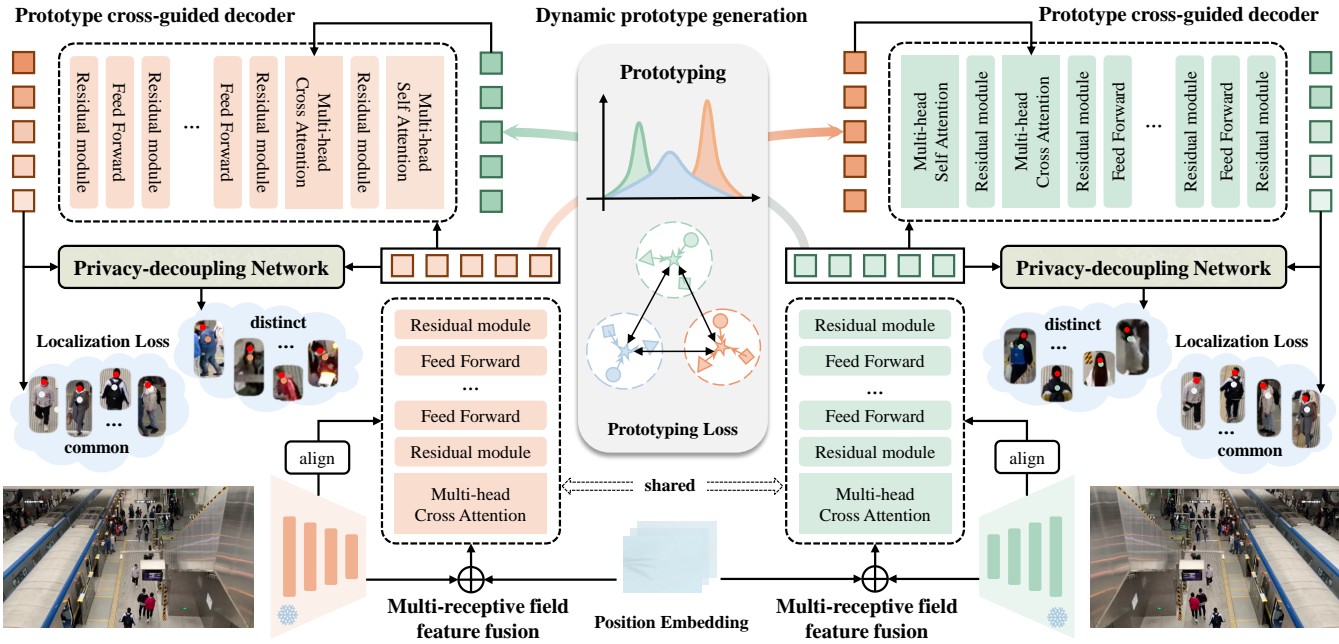

Figure 2: An overview of our proposed PDTR. Our model includes four main parts, i.e., multi-receptive field feature fusion (MRF³), dynamic prototype generation (DPG), prototype cross-guided decoder (PCD) and privacy-decoupling module (PDM). MRF³ integrates multi-scale and global information through an extractor and an encoder to generate multi-receptive field fusion features. DPG learns and extracts semantic consistency representations via clustering similar features across adjacent frames. PCD employs comprehensive features and prototypes to cross-guide a decoder for generating shared representations. PDM distinguishes private features from comprehensive ones to facilitate coordinate prediction.

Moreover, researchers have explored the potential of transformers in VCC. Fang et al. [9] thoroughly investigated the complementarity between density maps of consecutive frames, and addressed pedestrian occlusion in videos. Inspired by those methods, we first attempt to apply transformer to VIC and design a dual-transformer framework to handle temporal information in videos.

## 3 METHOD

### 3.1 Problem Formulation

Suppose that we have a video $I = \{I_i\}_{i=0}^{T-1}$ containing $T$ frames. Given the similarity between adjacent frames, it is possible that the flow of people has remained constant due to minor variations in information. Thus, we sample every $\varepsilon$ frames to create frame pairs $\{I_z, I_{z+\varepsilon}\}$. For each frame in $\{I_z, I_{z+\varepsilon}\}$, taking $I_z$ as an example, its corresponding label is $L_z = \{c_n^z, s_n^z\}_{n=1}^{N_z}$. Specifically, $c_n^z = (x_n^z, y_n^z)$ represents the coordinates of all $N_i$ pedestrians in frame $I_z$, while $s_n^z \in \{0, 1\}$ denotes corresponding state of pedestrian $n$. Here, 0 indicates the pedestrian appears in both frames, and 1 denotes that it only exists in frame $I_z$. The objective of this paper is to accomplish video individual counting (VIC), which involves accurately determining the total number of individuals in video $I$ while ensuring no duplication in the count.

Building upon the works of [12, 26], VIC is broken down into two sub-problems: counting the total number of people $M(I_0)$ in the initial frame, and inferring the number of new individuals $M_{in}(I_z, I_{z+\varepsilon})$ who appear in subsequent frames. The total number of people in

video $I$ can be calculated:
$$Total \approx M(I_0) + M_{in}(I_z, I_{z+\varepsilon})$$
$$= count(c^0) + \sum_{o=0}^{o=(T-\varepsilon)/\varepsilon} count(\{c^{z+\varepsilon}|s^{z+\varepsilon} = 1, z = o\varepsilon\}),$$

where $count(\cdot)$ is the count operation and $\varepsilon$ denotes the sample interval.

### 3.2 Overview of PDTR

Based on the analysis in Sec. 3.1, we have designed an end-to-end regression network, named PDTR, which can directly predict the point coordinates of pedestrians belonging to different states. For ease of understanding, we will use superscripts $(*)$ in the following description to uniformly represent different frames. Specifically, as shown in Fig. 2, PDTR is composed of four components: (a) MRF³, which includes a feature extractor (backbone) $\mathcal{F}_{\theta_1}$ and a fusion encoder $\mathcal{F}_{\theta_2}$. Extractor $\mathcal{F}_{\theta_1}$ extracts multi-scale feature from input frames $I^1$ and $I^2 \in \mathcal{R}^{H \times W \times 3}$ respectively. Encoder $\mathcal{F}_{\theta_2}$ captures long-distance global information and integrates them into extracted features to yield multi-receptive field fusion representations $F^{(*)} \in \mathcal{R}^{h \times w \times c}$. (b) DPG, which learns and stores shared prototypical representations $\bar{X}^{(*)}$ from different frames. In this paper, $k$ prototypes $X = \{x_1, ..., x_k\} \in \mathcal{R}^{k \times c}$ are learned from features $F^{(*)}$ via EM mechanism, to excavate consistent representations of the same target across neighboring frames. We expect that these learned prototypes can assist the model in tackling challenges arising from scene

diversity. (c) PCD, which leverages the learned $F^{(*)}$ and $\bar{X}^{(*)}$ to generate shared feature $Z_c^{(*)} \in \mathcal{R}^{h \times w \times c}$ in a cross guidance manner, i.e., $\mathcal{U}_\vartheta : ((F^1, \bar{X}^2), (F^2, \bar{X}^1)) \rightarrow Z_c^{(*)}$. (d) PDM, which decouples discriminative and private features contained in different frames from comprehensive representations: $\mathcal{V}_\psi : (F^{(*)}, Z_c^{(*)}) \rightarrow Z_s^{(*)}$. Finally, a decoder $\mathcal{D}_\phi$ is adopted to map the learned features to final point coordinates and confidence scores $\mathcal{A}$, i.e., $\mathcal{D}_\phi : (Z_c^{(*)}, Z_s^{(*)}) \rightarrow \mathcal{A}$. It is noteworthy that our model, configured as a dual-stream transformer for neighboring frames, shares weights across components since each branch executes identical functions.

## 3.3 Multi-receptive Field Feature Fusion

Existing studies [21, 22] have illustrated a direct approach to capture fine-grained image features for overcoming the limitation of relying solely on local information. They employ neural networks to extract detailed information, and then integrate them into the transformer encoder to model pixel-wise spatial dependencies. However, these approaches overlook the irrationality of feature extraction at the same scale, as objects of various sizes and distances have different scales. To this end, we incorporate multi-scale information into transformer reasoning to comprehensively capture features of targets with different sizes, as depicted in Fig. 3(a). By involving multi-receptive field information, our model effectively identifies target regions targets with various sizes.

Specifically, we design MRF³ to fuse multi-scale and global context information, comprising two main components: feature extractor $\mathcal{F}_{\theta_1}$ and fusion encoder $\mathcal{F}_{\theta_2}$. The former enhances the receptive field by progressively deepening network layers, offering diverse multi-scale information, whereas the latter emphasizes global context and integrates varied spatial details. Initially, extractor $\mathcal{F}_{\theta_1}$ takes a pair of adjacent frames $I^{(*)} \in \mathcal{R}^{H \times W \times 3}$ as inputs, generating $c$-dimensional multi-scale feature sets $\left\{ \check{F}_1, \check{F}_2, \check{F}_3, \check{F}_4 \right\}^{(*)}$ with resolutions of 1/32, 1/16, 1/8 and 1/4 respectively. Subsequently, MRF³ employs a feature alignment pyramid to align embeddings of various scales $\check{F}_i^{(*)}$ (excluding $\check{F}_4^{(*)}$) to $\bar{F}_i^{(*)}$ of the same resolution. Different layers of feature alignment pyramid incorporate varying numbers of feature alignment (FA) blocks, according to the feature size. The specific configuration is outlined below:

$$\bar{F}_i^{(*)} = i \times FA(\check{F}_i^{(*)}; \theta_{FA_i}), i = \{1, 2, 3\}, \tag{1}$$

where the FA block comprises an average pooling layer and a convolution layer, and $\theta_{FA_i}$ represents trainable parameters of the $i$-th FA block. Finally, the encoder $\mathcal{F}_{\theta_2}$ adopts $H$ transformer blocks to comprehensively integrate feature representations of varying scales:

$$\begin{aligned} Q_h &= \bar{F}_h^{(*)} \cdot W_h^Q, K_h = \check{F}_4^{(*)} \cdot W_h^K, V_h = \check{F}_4^{(*)} \cdot W_h^V, \\ Head_h &= Attention(Q_h, K_h, V_h), \\ MH(Q, K, V) &= Concat(Head_1, ..., Head_H), \end{aligned} \tag{2}$$

where $Q_h$, $K_h$ and $V_h$ denote learnable parameters for the $h$-th head, and $Q_h$, $K_h$ and $V_h$ denote *query*, *key* and *value*, respectively. $H = 8$ heads are used in our implementation. It is important to highlight that multi-scale features are integrated solely into the initial three encoder layers, and subsequent blocks maintain self-attention mechanism. By employing cross attention mechanism, we

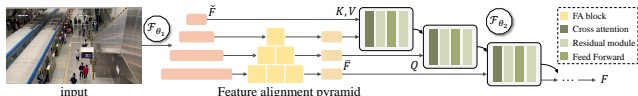

(a) Multi-receptive field feature fusion module

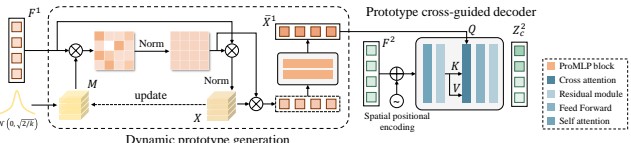

(b) Dynamic prototype generation and prototype cross-guided decoder

Figure 3: Illustration of MRF³ (a), DPG and PCD (b). MRF³ captures comprehensive features while DPG utilizes memory to generate prototypes that contain similar information. PCD reasons over prototypes and features to learn shared representations for VIC.

obtain fusion features $F^{(*)}$ that amalgamate rich spatial information from multiple receptive fields.

## 3.4 Dynamic Prototype Generation

Effectively utilizing similar information between adjacent frames to solve duplicate counting is a key insight in the VIC tasks. To delve deeper into the consistency information across frames in varied scenes, it is crucial to thoroughly consider the high-level semantic details within the scene. However, for the task of counting individuals in videos, traditional techniques like global average pooling are unreliable due to object-to-image issues and the dynamic nature of targets' motion [51]. Consequently, we employ metric learning to craft a dynamic prototype generation module (DPG), denoted as $\mathcal{P}_\delta$, to capture discriminative prototypical representations $\bar{X}^{(*)}$, as shown in Fig. 3(b). Motivated by the expectation-maximization (EM) strategy [6], DPG applies iterative clustering to dynamically group similar features and address motion-induced uncertainty. The feature inputs from different frames can then be expressed as the weighted sum of respective stored prototypes. This approach ensures the generation of semantic consistency representations for the same targets across neighboring frames.

To begin, reshape the given feature embedding $F^{(*)} \in \mathcal{R}^{h \times w \times c}$ into a set of local features by collapsing the spatial dimension of $F^{(*)}$ to one dimension, denoted as $F^{(*)} = \left\{ f_i^{(*)} \right\}_1^{hw}$, where $f_i^{(*)} \in \mathcal{R}^c$. Next, establish a collection of $k$ learnable visual atoms stored in an external memory as $M = \left\{ m_j \right\}_1^k$. To continually enhance memory performance, we initially sample from a normal distribution with a mean of 0 and a standard deviation of $\sqrt{2/k}$ to initialize the prototype embedding, i.e., $m_j \sim \mathcal{N}(0, \sqrt{2/k})$, and calculate the correlation map $C$ by:

$$C_{i,j} = \frac{e^{\rho m_j^T f_i^{(*)}}}{\sum_{i=1}^{hw} e^{\rho m_j^T f_i^{(*)}}}, \tag{3}$$

where $f_i^{(*)} \in F^{(*)}$ indicates the $i$-th feature, $m_j \in M$ denotes the $j$-th memory item, and $\rho$ represents an adjusting parameter. In

order to better fit the sample distribution in the input feature space, we update the memory $M$ based on the correlation map between samples and memory items calculated above:

$$\hat{x}_j = \frac{\sum_{i=1}^{hw} C_{i,j} f_i^{(*)}}{\sum_{i=1}^{hw} C_{i,j}}. \tag{4}$$

We then normalize the updated vector $\hat{x}_j$ to maintain unit length, which ensures placement in a standardized space, and contributes to model convergence and generalization:

$$x_j = \frac{\hat{x}_j}{\epsilon + \sqrt{\sum_{j=1}^{k} \hat{x}_j^2}}, \tag{5}$$

where $\epsilon = 1e - 6$, serves as a coefficient to prevent the denominator from being 0.

Followed by multiple iterations of the operations described above, the initial cluster centroids $X = \{x_1, ..., x_k\} \in \mathcal{R}^{k \times c}$ that have adjusted to the input feature distribution are acquired. To enhance the discriminative ability of these centroids, we introduce prototyping multi-layer perceptron (ProMLP) blocks to enhance representation $\bar{X}^{(*)}$ for individual frames:

$$\bar{x}_j^{(*)} = ProMLP(x_j; \theta_{mlp}^{(*)}), \tag{6}$$

where ProMLP consists of a stack of N = 2 identical MLP with residual connections, and $\theta_{mlp}$ represents the trainable parameters. Note that ProMLP is the only part of PDTR that does not share weights.

Ultimately, DPG generates the prototypical representations $\bar{X}^{(*)}$, capturing the similar semantics across various frames. In accordance with [44], an unsupervised prototyping loss is introduced to enforce a significant separation between the acquired prototypes:

$$\mathcal{L}_{pro} = \sum_{\bar{x}_i, \bar{x}_j \in \bar{X}^{(*)}} max((w - \left\| \bar{x}_i, \bar{x}_j \right\|_2^2), 0), \tag{7}$$

where $w$ is a pre-set hyperparameter.

## 3.5 Prototype Cross-guided Decoder

The prototype representations acquired through DPG contain rich inter-frame consistency information, which help model infer pedestrians that exist in both frames. To investigate shared information across adjacent frames utilizing the extracted features $F^{(*)}$ and prototype $\bar{X}^{(*)}$, we develop a prototype cross-guided decoder (PCD) denoted as $\mathcal{U}_\vartheta$. The decoder fully combines all available information and produces unified representations, i.e., $\mathcal{U}_\vartheta : ((F^1, \bar{X}^2), (F^2, \bar{X}^1)) \rightarrow (Z_c^1, Z_c^2)$. Leveraging the effectiveness of transformers in modeling long-distance dependencies, PCD employs a standard transformer architecture to capture inter-frame similarity and global information, as shown in Fig. 3(b). In contrast to prior methods [9, 22] that focused on learning close relationships within patches of single frame, we implement a cross-guidance strategy using prototypes from different frames to emphasize common regions. For frame $I^1$ in PCD, multi-head cross-attention mechanism computes the *query* using the prototype $\bar{X}^2$ from the current frame $I^2$ and derives the

*key* and *value* from the multi-scale feature $F^1$:

$$\tilde{Q}_h = \bar{X}^2 \cdot \tilde{W}_h^Q, \tilde{K}_h = F^1 \cdot \tilde{W}_h^K, \tilde{V}_h = F^1 \cdot \tilde{W}_h^V,$$
$$\tilde{Head}_h = Attention(\tilde{Q}_h, \tilde{K}_h, \tilde{V}_h), \tag{8}$$
$$\tilde{MH}(Q, K, V) = Concat(\tilde{Head}_1, ..., \tilde{Head}_H),$$

where $\tilde{Q}_h$, $\tilde{K}_h$ and $\tilde{V}_h$ represent the learnable parameters of project layers corresponding to the $h$-th head. Additionally, we also make the same cross-attention operation on the current frame $I^2$ utilizing information from previous frame $I^1$. In a word, the semantic information from various frames can be amalgamated into the shared features $Z_c^1$ and $Z_c^2$ through a prototypes cross-guided cross-attention operation. This strategy enables model to focus more effectively on similar regions and maintain semantic consistency.

## 3.6 Privacy-decoupling Module

Following the inference of shared pedestrians between two frames, it is necessary to additionally separate private features from comprehensive representation to infer pedestrians that only exist in their respective frames. To distinguish their distinct private domain information, we developed a privacy-decoupling module (PDM) $\mathcal{V}_\psi$ for decoupling comprehensive feature $F^{(*)}$ and shared features $Z_c^{(*)}$, sufficiently examining the disparities between two frames, i.e., $\mathcal{V}_\psi : (F^{(*)}, Z_c^{(*)}) \rightarrow Z_s^{(*)}$. Specifically, we assess spatial positional variances through feature subtraction, represented as:

$$Z_s^{(*)} = norm(F^{(*)} - drop(Z_c^{(*)})), \tag{9}$$

where $norm(\cdot)$ is LayerNorm and $drop(\cdot)$ is Dropout [41].

At this point, the model outputs shared features $Z_c^{(*)}$ and private features $Z_s^{(*)}$ from adjacent frames, which are used to predict the point coordinates (regression head) and confidence scores (classification head). A decoder $\mathcal{D}_\phi$ uses a simple MLP layer and a linear projection layer to achieve coordinate prediction and confidence scoring, respectively, i.e, $\mathcal{D}_\phi : (Z_c^{(*)}, Z_s^{(*)}) \rightarrow \mathcal{A}$.

To address the problem of ambiguous matching between adjacent targets in the video, we utilize KMO-based Hungarian bipartite matching [22] to guarantee the quality and rationality of the matching outcomes. The proposed PDTR is optimized using multiple loss functions, namely prototyping loss, regression loss, and classification loss. The first loss function has been defined in Section 3.4. The regression loss $\mathcal{L}_{reg}$ aims to regulate the learning of coordinate positioning and we employ the commonly-used *MAE* loss:

$$\mathcal{L}_{reg} = \left\| \mathcal{A}_c^{(*)} - \hat{\mathcal{A}}_c^{(*)} \right\|_1 + \left\| \mathcal{A}_s^{(*)} - \hat{\mathcal{A}}_s^{(*)} \right\|_1 \tag{10}$$

where $\hat{\mathcal{A}}_c^{(*)}$ and $\hat{\mathcal{A}}_s^{(*)}$ represent the true coordinate sets of shared and private targets, respectively. In addition, $\mathcal{A}_c^{(*)}$ and $\mathcal{A}_s^{(*)}$ denote respective predicted subsets obtained through the KMO-based Hungarian bipartite matching.

Moreover, we employ the focal loss [25] as the classification loss $\mathcal{L}_{cls}$ to distinguish whether predicted point belongs to key target or "background". The overall loss function of PDTR is:

$$\mathcal{L}_{PDTR} = \mathcal{L}_{pro} + \alpha_1 \mathcal{L}_{reg} + \alpha_2 \mathcal{L}_{cls}, \tag{11}$$

where $\alpha_1$ and $\alpha_2$ are scaling factors.

Table 1: Counting performance on CroHD dataset. CroHD11−CroHD15 are five test videos, and 133−321 correspond to ground truth. '-' denotes results that are unavailable due to unreleased code. '(·)' means the difference between the prediction and ground truth. 'T', 'L', 'M' and 'R' represent methods based on MOT, cross-line, feature matching and regression, respectively. '↓' indicates that the lower the better. The best and second-best results are highlighted in red and blue fonts.

| Methods | Venue | Key | MAE↓ | MSE↓ | RMAE(%)↓ | Counting results in five testing scenes | | | | |
| | | | | | | CroHD11 133 | CroHD12 737 | CroHD13 734 | CroHD14 1040 | CroHD15 321 |
|---|---|---|---|---|---|---|---|---|---|---|
| PHDTT[36] | IW-FCV 2022 | | 2130.4 | 2808.3 | 401.6 | 380 (247) | 4530 (3793) | 5528 (4794) | 1531 (491) | 1648 (1327) |
| FairMOT[45] | IJCV 2021 | T | 256.2 | 300.8 | 44.1 | 144 (11) | 1164 (427) | 1018 (284) | 632 (408) | 472 (151) |
| HeadHunter-T[34] | CVPR 2021 | | 253.2 | 351.7 | 32.7 | 198 (65) | 636 (101) | 219 (515) | 458 (582) | 324 (3) |
| LOI[47] | ECCV 2016 | L | 305.0 | 371.1 | 46.0 | 72.4 (60) | 493.1 (243) | 275.3 (458) | 409.2 (630) | 189.9 (131) |
| DRNet[12] | CVPR 2022 | M | 141.1 | 192.3 | 27.4 | 164.6 (31) | 1075.5 (338) | 752.8 (18) | 784.5 (255) | 382.3 (61) |
| CGNet[26] | CVPR 2024 | | 75.0 | 95.1 | 14.5 | - (7) | - (72) | - (14) | - (144) | - (138) |
| PDTR (Ours) | - | R | 60.6 | 73.7 | 12.7 | 109 (24) | 678 (59) | 729 (5) | 935 (105) | 431 (110) |

## 4 EXPERIMENTS

### 4.1 Experimental Settings

**Datasets.** Two publicly available benchmark datasets, namely CroHD [34] and SenseCrowd [19], are utilized for performance evaluation. Both datasets include annotations for point coordinates as well as inflow and outflow statistics. CroHD comprises 11,463 frames distributed over 9 full-HD resolution sequences, with 4 videos allocated for training and validation, and 5 videos designated for testing. SenseCrowd, a large-scale dataset with various scenes categorizations (e.g., density, time, and space), encompasses 634 sequences totaling 62,938 frames. The distribution of training, validation, and testing sets aligns with the experimental setups in [12].

**Evaluation Metric.** We uses Mean Absolute Error (MAE), Mean Square Error (MSE) and Weighted Relative Absolute Errors (WRAE) to evaluate performance. The first two metrics represent fundamental measures for crowd counting [46]. Additionally, in line with [12], we utilize WRAE to precisely assess the variance between video length and pedestrian count:

$$WRAE = \sum_{i=1}^{K} \frac{T_i}{\sum_{j=1}^{K} T_j} \frac{|N_i - \hat{N}_i|}{N_i} \times 100\%, \qquad (12)$$

where $K$ signifies the total count of videos, $T_i$ denotes the overall number of frames in $i$-th video, while $N_i$ and $\hat{N}_i$ represent the actual and predicted pedestrian quantities in $i$-th video.

**Implementation Details.** During training, the underlying extractor $\mathcal{F}_{\theta_1}$ is initialized by ResNet-50 [13] pre-trained on ImageNet [17], and the remaining modules (i.e., encoder $\mathcal{F}_{\theta_2}$, DPG $\mathcal{P}_\delta$, transformer decoder $\mathcal{U}_\vartheta$ and decoder $\mathcal{D}_\phi$) are randomly initialized. Encoder $\mathcal{F}_{\theta_2}$ and transformer decoder $\mathcal{U}_\vartheta$ both consist of 3 transformer layer with 8 heads. The learning rate is set as $1e-4$ along with Adam [16] with weight decay $5e-4$. The proposed network is trained on the Nvidia RTX 3090 GPU and implemented using the PyTorch framework. The scaling factor of losses $\alpha_1$ and $\alpha_2$ are both set to 0.5. The sampling interval for image pairs ranges from 2s to 8s, while it is set to 3s during testing.

**Comparison methods.** Currently, there is limited research on VIC

tasks, DRNet [12] and GCNet [26] are the only two for VIC. We categorize such approaches as feature matching-based methods, labeled as M. Furthermore, to assess the efficacy of our method, we also compared it with two distinct types of approaches: those based on multi-object tracking and those based on cross-line counting. The tracking-based method employs statistical analysis of object trajectories, including PHDTT [36], FairMOT [45], and HeadHunter-T [34], denoted as T. The cross-line method, LOI [48], involves counting the individuals crossing predefined lines, denoted as L.

### 4.2 Experimental Results

**Results on CroHD.** In Table 1, we present a comparison of our PDTR with 6 existing methods using three commonly-used metrics. The results indicate that our PDTR outperforms all competitors by a significant margin. For example, MAE and MSE of PDTR are remarkably low at 60.6 and 73.7, representing approximately one-fourth and one-fifth of those obtained by the tracking-based and cross-line methods, respectively. At the same time, it significantly surpasses feature matching-based methods, which proves the importance of reasoning over the similarity and difference between adjacent frames. For specific performance on five testing scenes, our method achieves the best prediction results, with the exception of CroHD11 and CroHD15. We achieve similar prediction results to the state-of-the-art on CroHD11 and CroHD15, mainly due to diversity of perspectives in the scene. As PDTR is trained solely on data from a fixed perspective, it exhibits limited adaptability to variations in shooting angles. Furthermore, PDTR demonstrates outstanding performance on several other high-density videos (e.g., CroHD12: 183.0, CroHD13: 259.6, and CroHD14: 245.9 person/frame), underscoring the effectiveness of our method in managing scenes with high density.

**Results on SenseCrowd.** Table 2 presents the model's performance on the larger dataset, SenseCrowd. It is evident that our model has attained performance comparable to the state-of-the-art in three metrics. Additionally, PDTR has exhibited commendable adaptability and robustness across varying scene conditions, i.e., time and space. Moreover, as scene density increases, occlusion

**Table 2: Counting performance on SenseCrowd dataset. $D_0 - D_4$ represent five density ranges respectively: [0, 50), [50, 100), [100, 150), [150, 200) and [200, +∞). '-' denotes unavailable results due to unreleased code. '↓' indicates that the lower the better. The best and second-best results are highlighted in red and blue fonts.**

| Methods | Venue | Key | MAE↓ | MSE↓ | RMAE(%)↓ | Density levels | | | | | Time | | Space | |
|---|---|---|---|---|---|---|---|---|---|---|---|---|---|---|
| | | | | | | $D_0$ | $D_1$ | $D_2$ | $D_3$ | $D_4$ | Day | Night | Indoor | Outdoor |
| FairMOT[45] | IJCV 2021 | T | 35.4 | 62.3 | 48.9 | 13.5 | 22.4 | 67.9 | 84.4 | 145.8 | 27.3 | 35.6 | 27.7 | 34.9 |
| HeadHunter-T[34] | CVPR 2021 | T | 30.0 | 50.6 | 38.6 | 11.8 | 25.7 | 56.0 | 92.6 | 131.4 | 29.2 | 32.8 | 31.7 | 29.5 |
| LOI[47] | ECCV 2016 | L | 24.7 | 33.1 | 37.4 | 12.5 | 25.4 | 39.3 | 39.6 | 86.7 | 26.8 | 17.8 | 22.6 | 25.4 |
| DRNet[12] | CVPR 2022 | M | 12.3 | 24.7 | 12.7 | 4.1 | 8.0 | 23.3 | 50.0 | 77.0 | 11.8 | 14.1 | 12.6 | 12.2 |
| CGNet[26] | CVPR 2024 | M | 8.9 | 17.7 | 12.6 | 5.0 | 5.8 | 8.5 | 25.0 | 63.4 | - | - | - | - |
| PDTR (Ours) | - | R | 9.6 | 17.6 | 11.4 | 4.6 | 6.8 | 14.7 | 23.6 | 60.6 | 10.5 | 13.5 | 10.5 | 10.4 |

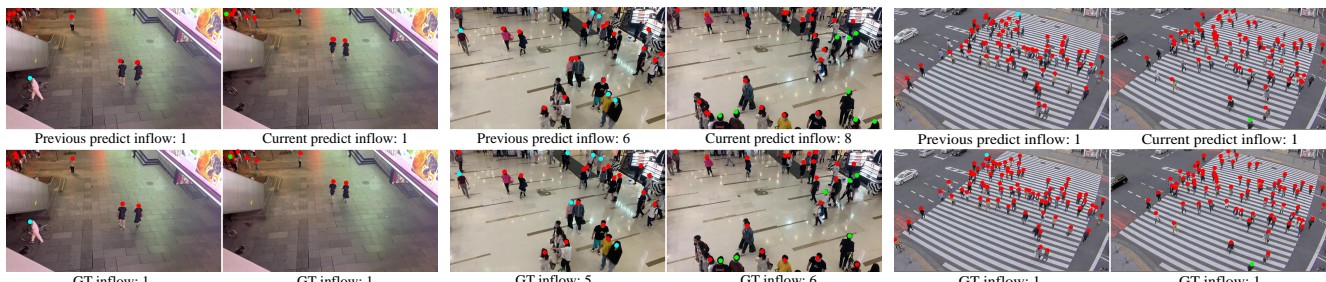

**Figure 4: Results on the SenseCrowd dataset. The top/bottom rows are predictions and ground truth, respectively. Red/light blue/green dots represent pedestrians present in both frames, outflow and inflow, respectively.**

problems become more prevalent, which leads to insufficient availability of feature information. This hampers the fulfillment of detection and association operations as necessitated by alternative methods, ultimately resulting in suboptimal model performance. On the contrary, our method's performance advantage becomes increasingly evident, as it directly converts input into coordinate points by leveraging the similarity and difference between two frames. Fig. 4 displays qualitative results of three representative scenarios (i.e., square, mall and street), demonstrating our model's ability to achieve accurate inference across various environments. Nonetheless, some erroneous examples may arise, as observed in current frame of second scene, where predicted inflow exceeds true inflow. Upon comparing adjacent frames, it becomes evident that PDTR provides more plausible prediction results.

### 4.3 Ablation study

**Impact of FAP.** The variety of perspectives across different scenarios result in pedestrian targets suffer from inconsistent scales. To assess the effectiveness of FAP developed in this study, we excluded this module from the original framework. Instead, we directly employed features obtained from extractor $\mathcal{F}_{\theta_1}$ as input for encoder $\mathcal{F}_{\theta_2}$ to generate comprehensive representations, as shown in Table 3 (i.e., w/o FAP). The counting performance exhibits a notable decline, particularly in CroHD13 and CroHD14 where scale issues are more prominent. Thus, the effectiveness and robustness of feature alignment pyramid in integrating multi-scale features and global information have been confirmed.

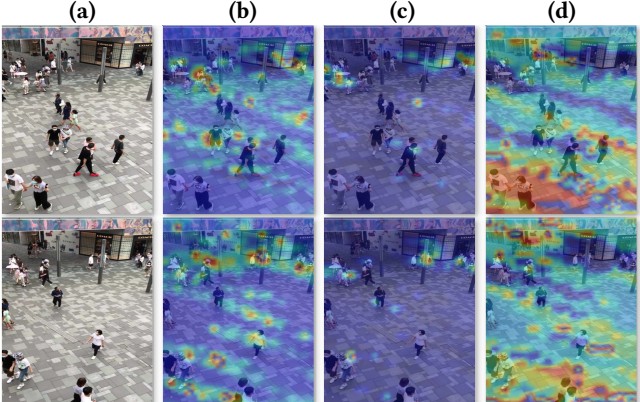

**Figure 5: Learned prototypes visualization for similar information on SenseCrowd dataset. The top/bottom rows represent previous and current frames, respectively. (a) Frames; (b)-(d) Visualizations of prototypes from different groups. Notably, (b) focuses on pedestrians; (c)-(d) pay attention to same targets between adjacent frames.**

**Impact of DPG.** Similarity information between neighboring frames is pivotal in addressing duplicate counting, guiding decoding and generating shared features. To verify the effectiveness of dynamic prototype generation devised in this study, we remove it from the overall model and directly utilize the outputs $\mathcal{F}^1$ and $\mathcal{F}^2$ of

**Table 3: Ablation studies on CroHD datasets. 'w/o FAP', 'w/o DPG', 'w/o $\mathcal{L}_{pro}$' and 'w/o Cross-guide' indicate that removing feature alignment pyramid, dynamic prototype generation, prototyping loss and cross-guidance mode from the overall model, respectively. '↓' indicates that the lower the better. The best results are highlighted in red font.**

| Methods | MAE↓ | MSE↓ | RMAE(%)↓ | Counting results in five testing scenes | | | | |
|---|---|---|---|---|---|---|---|---|
| | | | | CroHD11 133 | CroHD12 737 | CroHD13 734 | CroHD14 1040 | CroHD15 321 |
| w/o FAP | 117.2 | 149.7 | 18.0 | 195 (62) | 718 (19) | 989 (255) | 839 (201) | 272 (49) |
| w/o DPG | 99.2 | 119.7 | 15.5 | 170 (37) | 674 (63) | 868 (134) | 1255 (215) | 274 (47) |
| w/o $\mathcal{L}_{pro}$ | 90.8 | 123.5 | 13.2 | 180 (47) | 758 (21) | 842 (108) | 793 (247) | 290 (31) |
| w/o Cross-Guide | 141.6 | 163.2 | 23.0 | 181 (48) | 622 (115) | 967 (233) | 1282 (242) | 251 (70) |
| PDTR (Ours) | 60.6 | 73.7 | 12.7 | 109 (24) | 678 (59) | 729 (5) | 935 (105) | 431 (110) |

**Table 4: Impact of parameter settings on CroHD dataset. 'L' denotes transformer size (i.e., the number of layers), and '$k$' is the number of learnable visual atoms. '↓' indicates that the lower the better. The best results are highlighted in red font.**

| Settings | | | | | Counting results in five testing scenes | | | | |
|---|---|---|---|---|---|---|---|---|---|
| L | $k$ | MAE↓ | MSE↓ | RMAE(%)↓ | CroHD11 133 | CroHD12 737 | CroHD13 734 | CroHD14 1040 | CroHD15 321 |
| 3 | 128 | 75.6 | 82.9 | 15.7 | 177 (44) | 602 (135) | 790 (56) | 948 (92) | 372 (51) |
| 3 | 256 | 60.6 | 73.7 | 12.7 | 109 (24) | 678 (59) | 729 (5) | 935 (105) | 431 (110) |
| 3 | 512 | 86.4 | 110.29 | 16.4 | 145 (12) | 589 (148) | 741 (7) | 865 (175) | 411 (90) |
| 6 | 256 | 116.4 | 146.6 | 21.8 | 160 (27) | 578 (159) | 746 (12) | 786 (254) | 451 (130) |
| 9 | 256 | 123.8 | 174.6 | 19.9 | 139 (6) | 657 (80) | 761 (27) | 694 (346) | 481 (160) |

encoder $\mathcal{F}_{\theta_2}$ as "prototypes" to cross-guide decoder $\mathcal{U}_\vartheta$. The results are shown in Table 3 (i.e., w/o DPG). Evidently, the model without DPG experiences a significant decline compared to the fully-equipped model, suggesting that DPG enhances counting performance. Furthermore, we group prototypes and present visualizations of some groups in Fig. 5. It's evident that DPG effectively clusters semantically similar pixels, facilitating a more comprehensive understanding of same targets in different frames.

**Impact of $\mathcal{L}_{pro}$.** The prototyping loss is designed to enhance the discriminative ability of prototypical representations obtained from the memory module by maximizing their inter-distance. We remove it from the overall model to illustrate its role in constraining prototype generation, as shown in Table 3 (i.e., w/o $\mathcal{L}_{pro}$). Clearly, the model trained without prototyping loss function exhibits a notable performance decrease compared to the full-equipped model. Additionally, Table 4 demonstrates that setting $k$=256 visual atoms empowers our model to achieve outstanding performance.

**Impact of Cross-guide.** Effectively fusing information from adjacent frames using cross-guidance mode is another crucial aspect of achieving rational reasoning. To further clarify the role of cross-guidance mode in aiding generation of shared representations, we omit it and instead utilize outputs of encoder $\mathcal{F}_{\theta_2}$ to direct the decoding processes of each frame, as depicted in Table 3 (w/o Cross-Guide). Without cross-guidance mode, the model regresses into a single-frame counting model, lacking the ability to supplement information from adjacent frames and incapable of addressing inter-frame similarities and differences, consequently leading to a substantial decline in model performance.

**Impact of Transformer.** Increasing the number of transformer layer can enhance the model's representational capacity and more effectively capture the intricate relationships within input sequences [10, 35]. However, the excessive number of transformer layers may inflate computational costs and parameter numbers, increasing the difficulty of model optimization. We examine the impact of the encoder/decoder layer number on counting performance, as shown in Table 4. It is evident that with increasing layers, network performance decreases. At L=3, the network's counting performance peaks and generally surpasses that of higher layer counts.

## 5 CONCLUSION

In this paper, we presented a prototype-guided dual transformer reasoning (PDTR) framework to solve video individual counting (VIC) in an end-to-end regression manner. PDTR fully leveraged the advantages of prototype learning and dual-stream inference, innovatively using transformer to capture similarity and differences between adjacent frames for accurate counting. Initially, a multi-receptive field feature fusion module (MRF[3]) learned comprehensive representations of various frames. Subsequently, a dynamic prototype generation module (DPG) extracted similar information from these representations to cross-guide decoder (PCD) for shared features. Finally, a privacy-decoupling module (PDM) was implemented to extract frame-specific private information from comprehensive features. Extensive experiments conducted on two public datasets showed promising performance compared to state-of-the-art methods on VIC.

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
