# OpenReview forum: "Prototype-Guided Dual-Transformer Reasoning for Video Individual Counting"
_acmmm.org/ACMMM/2024/Conference — MM2024 Poster_

### Official Review · Reviewer_J1cf · 2024-06-01

**Rating:** 4
**Confidence:** 3

**Summary:**

The paper introduces a novel framework named Prototype-Guided Dual-Transformer Reasoning (PDTR) aimed at addressing the challenge of accurately counting individuals in video footage. This task, known as Video Individual Counting (VIC), is essential for managing densely populated urban areas and public spaces.

**Strengths:**

1. The authors propose a new framework that uses prototype learning and dual-stream inference to count individuals in a video. The framework is designed to handle the challenges of target repetition and motion state continuity in videos by converting the feature matching process into an end-to-end regression reasoning procedure.
2. The PDTR framework was evaluated on two public datasets, CroHD and SenseCrowd. The results consistently demonstrated the superiority of PDTR over state-of-the-art baselines, showcasing its effectiveness in challenging scenarios with occlusions, dynamic traffic, and complex layouts. Further, the authors conducted ablation studies to assess the contribution of each module within the PDTR framework.

**Limitations:**

1. The PDTR might be more complex compared to simpler counting methods, which could impact its ease of implementation and computational efficiency. The use of transformers and multiple loss functions might increase the computational cost, which could be a limitation for users with limited computational resources.
2. The underlying extractor is initialized with a pretrained model on ImageNet, which may introduce dependencies on external resources and the quality of pretrained weights.
3. The paper could benefit from a more detailed comparative analysis with existing methods, particularly in terms of computational efficiency and scalability.
4. While the paper demonstrates the effectiveness of the framework in controlled settings, its effectiveness in real-world applications and its integration into practical systems could be further explored.

**Suitability:**

2

---

### Official Review · Reviewer_Zg19 · 2024-06-03

**Rating:** 4
**Confidence:** 3

**Summary:**

The paper presents a novel framework, Prototype-Guided Dual-Transformer Reasoning (PDTR), for Video Individual Counting (VIC). The approach aims to accurately count the total number of individuals in a video without duplication. The PDTR framework leverages prototype learning and dual-stream inference to achieve this goal. It includes four main components: multi-receptive field feature fusion (MRF3), dynamic prototype generation (DPG), prototype cross-guided decoder (PCD), and privacy-decoupling module (PDM). Extensive experiments demonstrate that PDTR outperforms state-of-the-art baselines on existing VIC datasets.

**Strengths:**

- Innovative approach
- handling shared and private features can provide a better solution than current approaches
- e2e regression reasoning
- highly experimental and ablation studies

**Limitations:**

- to complex approach.
- paper lack conceptual innovation, but the components used looks more of an engineering effort
- If the paper mainly combines well-known techniques (e.g., transformers, prototype learning) in a new context (video individual counting), this combination might be viewed as evolutionary rather than revolutionary
- Yao, T., Li, Y., Pan, Y., Wang, Y., Zhang, X. P., & Mei, T. (2023). Dual vision transformer. IEEE transactions on pattern analysis and machine intelligence. This paper can be useful.

**Suitability:**

3

---

### Official Review · Reviewer_sErm · 2024-06-03

**Rating:** 4
**Confidence:** 4

**Summary:**

The paper introduces a Prototype-Guided Dual-Transformer Reasoning (PDTR) framework designed to count individuals in videos accurately without duplication. This method leverages a dual-transformer approach to capture similarities and differences between adjacent frames in an end-to-end regression manner. Key components include:

Multi-receptive Field Feature Fusion Module (MRF3): Acquires initial comprehensive representations.
Dynamic Prototype Generation (DPG): Generates consistent representations across frames.
Prototype Cross-Guided Decoder (PCD): Uses shared features for improved counting.
Privacy-Decoupling Module (PDM): Separates private features to enhance accuracy.
Extensive experiments on two VIC datasets demonstrate the PDTR's superior performance over existing methods.

**Strengths:**

S1: Innovative dual-transformer framework that effectively handles the similarities and differences between adjacent video frames for accurate individual counting.

S2: The dynamic prototype generation module enhances consistency in representations across frames, reducing duplicate counting.

S3: Extensive use of privacy-decoupling ensures that private features are well-separated, aiding in better individual identification.

S4: Comprehensive testing on multiple datasets shows robust performance across different scenarios.

S5: The multi-receptive field feature fusion module integrates multi-scale and global context information, improving feature representation.

**Limitations:**

W1: The framework is specialized for counting tasks and does not address other potential tasks like tracking or behavior analysis, limiting its broader applicability.

W2: The addition of multiple complex modules, such as DPG and PCD, increases computational requirements, potentially impacting real-time performance.

W3: While tested on various datasets, the method's performance in highly diverse and unforeseen environments is not thoroughly examined, raising questions about its generalizability.

Overall Review
The paper presents a novel and effective approach to video individual counting using a Prototype-Guided Dual-Transformer Reasoning framework. Its strengths lie in the innovative use of dual transformers and prototype generation to enhance counting accuracy while maintaining privacy. The method shows significant improvements over existing techniques, especially in handling frame-to-frame consistency and privacy concerns. However, the added complexity and computational demands could hinder real-time applications, and broader testing is needed to ensure its robustness in diverse settings. Overall, the PDTR framework is a promising advancement in the field of video individual counting.

**Suitability:**

3

---

### Meta-Review · Area_Chair_YzwK · 2024-06-30

**Recommendation:** Accept (Poster)
**Confidence:** 5

**Metareview:**

This work is focused on individual counting in videos. The authors propose a prototype guided approach to solve this problem. This work initially received 3x borderline accepts and the reviewers were mainly concerned by the complex nature of the approach. The authors provided a rebuttal and most of the concerns were addressed in the response. The final ratings are weak accept and 2x borderline accept. The AC agrees with this assessment and even though the solution is little complex it is performing well and the evaluation is comprehensive. The AC recommends acceptance of this work.